# Compromising the 19S proteasome complex protects cells from reduced flux through the proteasome

Peter Tsvetkov[1], Marc L Mendillo[2], Jinghui Zhao[3], Jan E Carette[4], Parker H Merrill[5], Domagoj Cikes[6], Malini Varadarajan[1], Ferdy R van Diemen[7], Josef M Penninger[6], Alfred L Goldberg[3], Thijn R Brummelkamp[7], Sandro Santagata[1,8,9]*, Susan Lindquist[1,2]*

[1]Whitehead Institute for Biomedical Research, Cambridge, United States; [2]Department of Biology, Howard Hughes Medical Institute, Massachusetts Institute of Technology, Cambridge, United States; [3]Department of Cell Biology, Harvard Medical School, Boston, United States; [4]Department of Microbiology and Immunology, Stanford University School of Medicine, Stanford, United States; [5]Department of Pathology, Brigham and Women's Hospital, Harvard Medical School, Boston, United States; [6]Institute of Molecular Biotechnology of the Austrian Academy of Sciences, Vienna, Austria; [7]Department of Biochemistry, Netherlands Cancer Institute, Amsterdam, Netherlands; [8]Department of Pathology, Brigham and Women's Hospital, Boston, United States; [9]Department of Cancer Biology, Dana-Farber Cancer Institute, Harvard Medical School, Boston, United States

*For correspondence: ssantagata@bics.bwh.harvard.edu (SS); lindquist_admin@wi.mit.edu (SL)

Competing interests: The authors declare that no competing interests exist.

**Abstract** Proteasomes are central regulators of protein homeostasis in eukaryotes. Proteasome function is vulnerable to environmental insults, cellular protein imbalance and targeted pharmaceuticals. Yet, mechanisms that cells deploy to counteract inhibition of this central regulator are little understood. To find such mechanisms, we reduced flux through the proteasome to the point of toxicity with specific inhibitors and performed genome-wide screens for mutations that allowed cells to survive. Counter to expectation, reducing expression of individual subunits of the proteasome's 19S regulatory complex increased survival. Strong 19S reduction was cytotoxic but modest reduction protected cells from inhibitors. Protection was accompanied by an increased ratio of 20S to 26S proteasomes, preservation of protein degradation capacity and reduced proteotoxic stress. While compromise of 19S function can have a fitness cost under basal conditions, it provided a powerful survival advantage when proteasome function was impaired. This means of rebalancing proteostasis is conserved from yeast to humans.

## Introduction

Maintaining the integrity of the proteome is vital for all cells. Protein chaperone systems and the ubiquitin-proteasome pathway are essential components of the global architecture that sustains protein homeostasis. The importance of these systems is underscored by the fact that perturbation of protein homeostasis is central to diverse human diseases (*Balch et al., 2008*; *Labbadia and Morimoto, 2015*). In neurodegenerative diseases, cells succumb to an overload of toxic protein aggregates. Whereas in cancer, malignant cells co-opt protein quality control systems to accommodate the severe proteotoxic stresses that arise from high mutation loads, relentless biomass accumulation and protein turnover (*Mendillo et al., 2012*; *Deshaies, 2014*; *Scherz-Shouval et al., 2014*).

Because the ubiquitin-proteasome system is the major mechanism regulating protein turnover, cells are highly vulnerable to insults that impair the function of this system (*Varshavsky, 2012*;

**eLife digest** Cells have numerous methods for removing proteins that have been damaged or are no longer needed. One of these methods is carried out by a large protein complex called the proteasome. Because of its central role in maintaining protein balance, drugs that stop the proteasome functioning often kill cancer cells grown in dishes. However, these proteasome inhibitors tend not to work against most tumors in patients. Moreover, tumors that do respond to these drugs ultimately become resistant to them.

Tsvetkov et al. used a genetic screen to find the mutations that allowed cancer cells to withstand exposure to proteasome inhibitors. The proteasome complex contains two types of subunits: regulatory subunits that recognize the proteins that need to be degraded; and catalytic subunits that degrade the proteins. Surprisingly, individually inactivating the genes for many different regulatory subunits provided protection against proteasome inhibitors.

When the regulatory subunits were reduced, the proteasomes shifted into a state that ultimately protected the cells. This mechanism was observed to protect both yeast and human cells and may be a widespread mechanism for establishing stress-resistant states.

The next challenge will be to identify the vulnerabilities of cells that have reduced regulatory subunits. Research is also needed to find out if this reduction varies from cell to cell, making some cells more able to withstand treatment.

Petrocca et al., 2013). Proteasome function can be hampered in many ways. For example, a multitude of environmental stresses cause the toxic accumulation of misfolded and aggregated proteins (Parsell et al., 1994; Gidalevitz et al., 2011). Such proteins can bind the proteasome and stabilize an inactive closed conformation (Deriziotis et al., 2011; Ayyadevara et al., 2015). In addition, chemically diverse natural product inhibitors of the ubiquitin-proteasome pathway are elaborated by organisms as diverse as terrestrial and marine bacteria, fungi, and plants (Kisselev et al., 2006; Schneekloth and Crews, 2011; Kisselev et al., 2012). These proteasome inhibitors can thwart the proliferation of neighboring organisms and thereby provide a competitive growth advantage.

Applying this lesson from nature toward the treatment of cancer has been an effective strategy. Malignant cells in which the rates of protein synthesis outstrip proteasome degradation capacity are exceedingly vulnerable to proteasome inhibition (Adams et al., 2000; Cenci et al., 2012; Deshaies, 2014). Targeting the 26S proteasome with the proteasome inhibitor bortezomib (Velcade, PS-341) is particularly effective in hematopoietic tumors. Indeed, it is broadly applied as a therapy for patients with myeloma or mantle cell lymphoma (Chen et al., 2011; Crawford and Irvine, 2013).

The 26S proteasome is an elaborate multi-subunit protein complex that is present in all eukaryotes. This complex is comprised of a 20S catalytic core that orchestrates peptide bond cleavage and a 19S regulatory complex that can be attached to either or both ends of the 20S core (Besche et al., 2009; Matyskiela et al., 2013; Tomko and Hochstrasser, 2013). The 19S complex recognizes ubiquitin-tagged substrates, cleaves ubiquitin chains, unfolds substrates and translocates the unfolded proteins into the catalytic chamber of the 20S core (Hochstrasser, 1996; Finley, 2009; Matyskiela et al., 2013).

Despite an exquisitely detailed understanding of proteasome function and of the mechanism of action of proteasome inhibitors, we have a limited understanding of the molecular mechanisms that cells deploy to resist the cytotoxic effects of reduced flux through the proteasome (Kale and Moore, 2012). Such an understanding is of great importance in the dynamics of natural ecosystems in the face of diverse proteotoxic stresses and in the clinic, where pre-existing intrinsic resistance and acquired resistance following drug exposure have limited the effectiveness of bortezomib as a therapeutic.

To gain insights into the mechanisms that allow cells to withstand reduced flux through the proteasome, we took advantage of highly specific chemical inhibitors. While proteasome function can be impaired by many factors, none can be controlled with the dosage-dependent precision of proteasome inhibitors such as bortezomib and the peptide aldehyde MG132. These inhibit both 20S and 26S proteasomes by targeting the core proteolytic catalytic activity of the 20S subunits (Kisselev et al., 2006, 2012; Goldberg, 2012).

We used these inhibitors at toxic levels in an unbiased, genome-wide screen. We selected for cells that were resistant to the inhibitors from a library of 100 million gene-trap insertions, using a human cell line that is haploid for all chromosomes except chromosome 8 (*Carette et al., 2009*, *2011a*). This approach is analogous to screens so broadly and effectively used in haploid yeast and identifies loss-of-function events that allow human cells to survive diverse toxic insults (*Guimaraes et al., 2011*; *Reiling et al., 2011*; *Carette et al., 2011a*, *2011b*; *Winter et al., 2014*). The results of our screen led us to the discovery of a surprising and highly conserved strategy by which organisms can protect themselves from the toxic effects of reduced flux through the proteasome.

## Results

### Unbiased mammalian screen identifies 19S subunits as key determinants of resistance to proteasome inhibitors

For our screens, we used a library of near-haploid human chronic myeloid leukemia cells (KBM7) containing approximately 100 million retroviral gene-trap insertions that target over 98% of transcribed genes. To identify genes that increase resistance to proteasome inhibition, we exposed cells for 4 weeks to either MG132 or bortezomib. We then further expanded the pools of resistant cells to enable the amplification and sequencing of the insertion sites (*Figure 1A*).

For the MG132 resistance screen, we identified 992 independent insertion sites in the pool of surviving cells. Surprisingly, all insertions that reached a high level of statistical enrichment (p-value $< 1 \, e^{-7}$) lay in genes encoding subunits of the proteasome 19S regulatory complex (*Figure 1B,C*; *Supplementary file 1*). These included both ATPase subunits (*PSMC2, PSMC3, PSMC4, PSMC5*, and *PSMC6*) as well as non-ATPase subunits (*PSMD2, PSMD6, PSMD7*, and *PSMD12*). No insertions were recovered in genes encoding subunits of the 20S catalytic core (*Supplementary file 1*). For the bortezomib resistance screen, we recovered 538 independent insertions. The results of the 2 screens were remarkably similar with seven of the ten most highly enriched genes encoding subunits of the 19S complex (*Supplementary file 1*).

Such strikingly similar results from 2 unbiased screens with chemically distinct proteasome inhibitors strongly indicated that altering the 19S complex can protect cells against compounds that inhibit the 20S catalytic core. From the resistant pools of cells, we then attempted to isolate stable clones that contained 19S subunit gene insertions. We were unable to do so. Next, we attempted to delete PSMD12 in a near-haploid fibroblast cell line (HAP1) (*Carette et al., 2010*; *Essletzbichler et al., 2014*). With this targeted approach using CRISPR constructs, we were only able to recover diploid cell variants in which just one of the two PSMD12 alleles was disrupted.

Finally, from a collection of haploid mouse embryonic stem cells that harbor reversible gene-trap cassettes (*Elling et al., 2011*), we identified two clones with cassettes located in the first intron of the *PSMC2* or *PSMD12* genes. In these cells, inversion of the cassettes would generally be expected to inactivate the targeted gene. We induced Cre-mediated inversion in over 3000 cells harboring each cassette, but less than 1% of the cells survived. We confirmed that inversion had occurred in the surviving cells. However, all of the stable clones that emerged retained expression of the targeted subunits (*Figure 1—figure supplement 1*).

These findings confirm that, as others have found in yeast and *Drosophila*, the function of the 19S regulatory complex is essential for sustained proliferation of mammalian cells under basal conditions. Presumably, cells carrying 19S mutations had become enriched in our initial MG132 and bortezomib screens because they provided cells with a short-term advantage for growth over several generations.

### Reducing 19S subunits protects human cancer cells from proteasome inhibitors

Next, we asked if a simple reduction in the expression of 19S subunits could protect against the toxicity of proteasome inhibitors. We assembled a panel of shRNA-expressing lentiviruses targeting seventeen 19S subunits and three 20S subunits (*PSMB5, PSMB7*, and *PSMA3*). Each gene was targeted separately using four different shRNAs (*Figure 2A*, *Supplementary file 2*) and we averaged the effects on viability from these four hairpins.

With or without bortezomib, knockdown of any of the subunits of the 20S catalytic core reduced viability (*Figure 2A*, *Figure 2—figure supplement 1A–D*). In contrast, the effects of knocking down several different 19S subunits had opposing effects, depending on the absence or presence of the inhibitor. In its absence, 19S subunit reduction had a fitness cost and decreased cell viability; in its

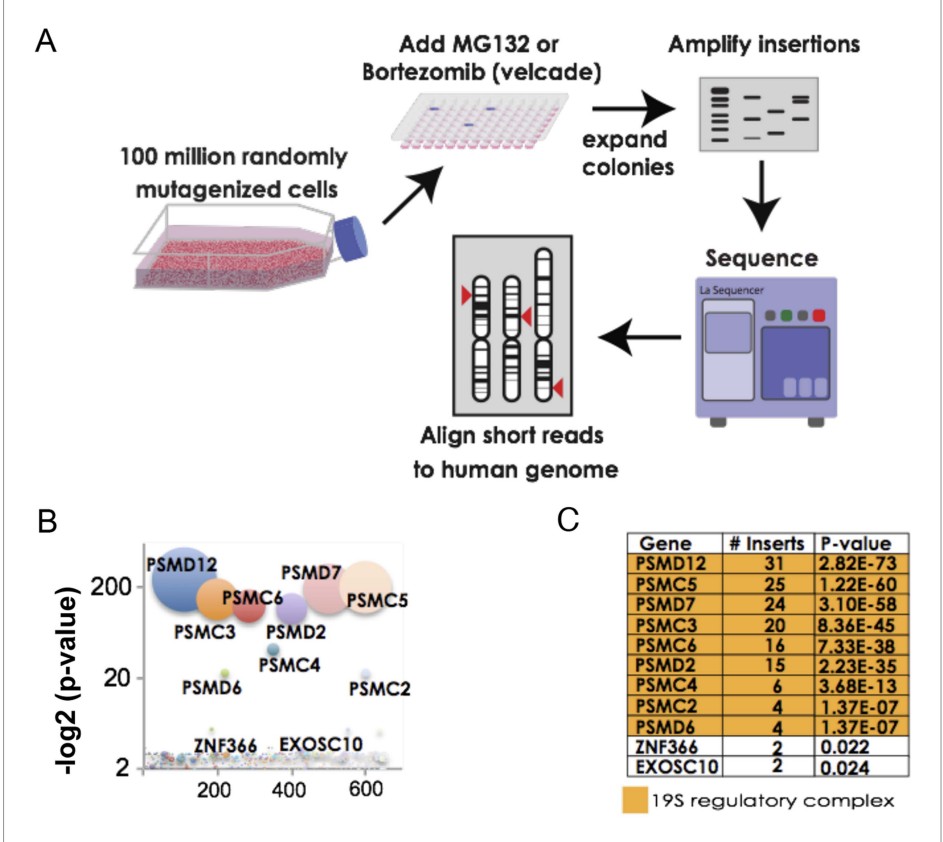

**Figure 1**. The 19S regulatory subunits of the proteasome are the most significant mediators of resistance to proteasome inhibitor toxicity. (**A**) Schematic representation of the screen. One hundred million KBM7 cells subjected to random gene deletion using retroviral gene-trap insertions were exposed to either MG132 (700 nM) or bortezomib (18 nM) for 4 weeks. Surviving cells were expanded and insertions identified by sequencing. (**B**) The p-values of the recovered insertions from the MG132 screen are plotted (log2). Bubble sizes represent the number of insertions. (**C**) Compilation of the most significant gene deletions conferring resistance to MG132 with the gene name, number of inserts, and p-value. The subunits of the 19S regulatory complex are highlighted with orange.

The following figure supplement is available for figure 1:

**Figure supplement 1**. (**A**) Schematic representation of the knockout strategy to generate the mutant PSMD12 and PSMC2 ES cells.

presence, 19S subunit reduction provided a survival advantage and increased viability (*Figure 2A*, *Figure 2—figure supplement 1A–D*).

To investigate further, we sought to recover stable clones with reduced levels of 19S subunits. Long-term 19S subunit reduction impeded the growth of most cells, but we were able to propagate two lines that proliferated normally (*Figure 2—figure supplement 1E*). These lines stably expressed shRNA targeting either PSMC5 or PSMD2. In both cases, the lines had only a modest reduction in protein levels (*Figure 2—figure supplement 1F*). At a concentration of bortezomib that completely inhibited the proliferation of control cells (12 nM), these cells continued proliferating (*Figure 2B*, *figure 2—figure supplement 1G*).

We next used native gels to assess the levels and ratios of 20S and 26S proteasome complexes following a 24-hr exposure to bortezomib (*Figure 2C*). Bortezomib-induced cytotoxicity was not observed in control cells at this time point (*Figure 2B*, star). In cells with reduced levels of either PSMC5 or PSMD2, 26S proteasome complexes were reduced and 20S proteasome complexes were increased (*Figure 2C*, upper panel). These changes persisted in the presence of bortezomib (*Figure 2C*). Next, we assayed 20S proteasome complex activity with a fluorescently labeled substrate

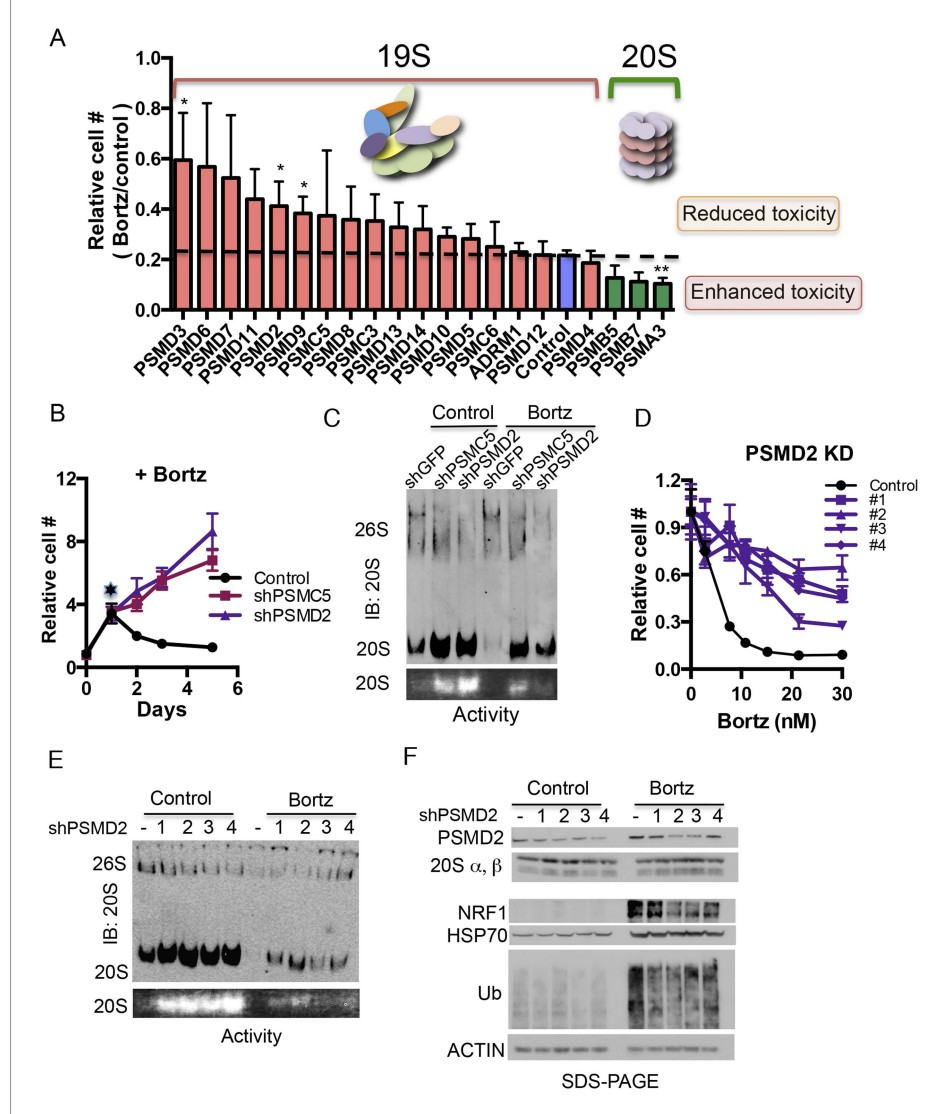

**Figure 2**. Reducing expression of 19S subunits increases the levels of active 20S proteasomes and protects cancer cells from proteasome inhibition. (**A**) HepG2 cells were infected with 80 shRNAs targeting 20 different subunits of the proteasome and 10 control shRNAs. Infected cells were then exposed to 12 nM bortezomib and cell number was examined 4 days later. The plot represents the average (± SEM) of four different hairpins targeting the indicated proteasome subunits and their relative cell number following bortezomib treatment. 19S subunits are depicted with orange bars, the control with a blue bar, and 20S subunits with green bars. (**B**) HepG2 cells harboring either a control shGFP or shRNAs targeting two proteasome subunits (shPSMC5, shPSMD2) displayed significant growth differences in the presence of 12 nM bortezomib. (**C**) Proteasome complex content in the shGFP-, shPSMC5-, and shPSMD2-expressing HepG2 cells was analyzed by native gel electrophoresis after 24 hr of treatment with or without 12 nM bortezomib revealing an increase in 20S proteasome levels and activity in cells knocked down for PSMC5 or PSMD2. (**D**) The relative cell number of cells harboring a control shLacZ or each of 4 individual shRNAs targeting shPSMD2 was analyzed 4 days after addition of the indicated concentrations of bortezomib. (**E**) HepG2 cells stably expressing four different shRNAs targeting the PSMD2 subunit and a control shRNA (lacZ) were analyzed by Western blot for the indicated proteins 24 hr with or without bortezomib treatment. (**F**) Proteasome complex levels and activity in a control HepG2 cells and 4 cell lines with reduced PSMD2 levels with and without a 24 hr incubation with bortezomib (12 nM). Proteasome complex levels were detected by immunoblot analysis and 20S proteasome activity by measuring the hydrolysis of Suc-LLVY-AMC by substrate overlay assays. In B and C, the graph represents the average of four replicas and their SEM. The p-values were obtained by conducting a two-tailed unpaired t-test. *p < 0.05 **p < 0.01 Bortz- Bortezomib (**E**).

*Figure 2. continued on next page*

*Figure 2. Continued*

The following figure supplement is available for figure 2:

**Figure supplement 1**. (**A–D**) Examining the effect of proteasome subunit knockdown in different cell lines.

on native gels. Exposing control cells to 12 nM bortezomib for 24 hr led to a nearly complete inactivation of 20S proteasome function (*Figure 2C*, lower panel). Cells with reduced levels of either PSMC5 or PSMD2 maintained a significant level of 20S proteasome activity (*Figure 2C*).

To validate our findings, we selected four additional cell lines each with a different shRNA driving modest reduction in PSMD2 protein levels. Again, all of these lines were significantly more resistant to bortezomib than the parental line (EC50 values increased by three to sixfold, *Figure 2D*). In these cells, we also observed that 26S proteasome complexes were reduced and that both the levels and the activity of 20S proteasome complexes were sharply increased (*Figure 2E*). Notably, in the presence of bortezomib, reducing 19S subunits enabled the relative preservation of the levels and activity of 20S proteasome complexes (*Figure 2E*).

## 19S subunit reduction does not activate classic cytoprotective stress responses and blunts bortezomib-mediated stress responses

A likely mechanism by which 19S subunit reduction might promote resistance is by induction of the cytoprotective stress responses that allow cells to cope with the increase in proteotoxic stress caused by the proteasome inhibitor. One major response to such inhibition is the activation of NRF1, a transcriptional regulator of proteasome gene expression that increases the expression of proteasome subunits, elevates proteasome content, and promotes resistance (*Radhakrishnan et al., 2010*; *Steffen et al., 2010*; *Radhakrishnan et al., 2014*; *Sha and Goldberg, 2014*). A second major response to proteasome inhibitors is activation of heat-shock factor 1 (HSF1), the master regulator of the heat-shock response, which increases levels of HSP70 and other protein chaperones.

Surprisingly, in the four cell lines with modest reductions of PSMD2, we did not detect constitutive activation of NRF1, and correspondingly, the expression of 20S subunits was unaltered (*Figure 2F*). Moreover, HSF1 was not activated as reflected by the stable expression of HSP70, the protein that is most highly responsive to proteotoxic stress (*Figure 2F*). Consistent with these findings, polyubiquitinated proteins did not accumulate in the PSMD2 knockdown cells (*Figure 2F*) suggesting that modestly reducing 19S subunit levels did not itself induce a cytoprotective stress response.

Not only were stress response pathways not activated, but the response to bortezomib was blunted in cells with reduced PSMD2 levels. The accumulation of polyubiquitinated proteins was reduced relative to control cells treated with the inhibitor. The activation of NRF1 was also reduced (*Figure 2F*). We obtained very similar results in cells with PSMC5 knockdown except in this case HSP70 levels were also reduced (*Figure 2—figure supplement 1H*).

Notably, the protective effect of 19S subunit reduction was specific to the toxicity caused by proteasome inhibitors. Cells with reduced PSMD2 levels remained fully sensitive to small molecule mediators of ER stress, HSP90 inhibition, thiol adduct formation or blockade of translation initiation or translation elongation among other stresses (*Figure 2—figure supplement 1I–N*). Thus, modest 19S subunit reduction protected cells by selectively lowering the proteotoxic stress that is generated by proteasome inhibition.

## Compromising the 19S regulatory complex suppresses bortezomib-induced stress responses

To examine in detail the transcriptional changes that characterize cells with increased resistance to proteasome inhibition, we performed whole-genome RNA-sequencing in two lines with modest reductions in PSMD2. Sequence data confirmed that PSMD2 mRNA levels were reduced by ∼50% in both lines. We examined basal gene expression and the effects of bortezomib treatment on these cells, comparing them to cells carrying a control lacZ shRNA construct.

Under basal conditions (in the absence of bortezomib), cells with reduction in PSMD2 showed a strong induction of components of the ribosome (gene set enrichment analysis false discovery rate [FDR]

q-value = 4.0 e$^{-22}$) (*Supplementary file 3*). Genes encoding the 20S subunits of the proteasome were not induced, consistent with our earlier observation that NRF1 is not activated and 20S subunit levels are unchanged (*Figure 2F*). The most strongly downregulated gene category involved components of diverse proteotoxic stress responses (FDR q-value = 2.31 e$^{-9}$) (*Supplementary file 3*). These included genes for sentinel proteins that respond to heat-shock (e.g., HSPA1A, HSPA1B, HSPA8, HSPB1, and HSP90AA1), oxidative stress (e.g., HMOX1), and ER stress (e.g., CHOP/DDIT, TRIB3, and HERPUD1) (*Figure 3A*). Genes previously identified as an HSF1-regulated cancer-specific transcriptional program were also downregulated (*Mendillo et al., 2012*; *Santagata et al., 2013*) (FDR q-value = 0.042; normalized enrichment score (NES) 1.57) (*Figure 3—figure supplement 1A*). Therefore, 19S subunit reduction not only fails to induce classic adaptive stress responses, but actually lowers basal levels of proteotoxic stress.

As expected, in control cells, bortezomib treatment unleashed a powerful transcriptional response characterized by a sharp increase in stress-response transcripts (FDR q-value = 7.07 e$^{-13}$) (*Supplementary file 3*). This potent bortezomib-induced stress response was markedly attenuated in cells with reduced PSMD2 levels (*Figure 3B*). The suppressed genes include ones involved in oxidative stress and ER stress responses, as well as genes in the HSF1-mediated heat-shock response (FDR q-value < 0.01; NES 2.317) (*Figure 3B,C* and *Figure 3—figure supplement 1B*). For example, induction of HSP70 family heat-shock genes HSPA1A and HSPA1B was reduced by eightfold, while HSPA6 was entirely suppressed (*Figure 3B*).

Previous work has identified 28 mediators of bortezomib toxicity, many of which are upregulated upon bortezomib exposure (*Chen et al., 2010*). When we treated control cells with bortezomib, the mRNA transcripts of many of these genes increased significantly (e.g., ATG4A, DDX27, GADD45A, NUP54, ODC1, PMAIP1/NOXA, SETX, SNIP1, and TAX1BP1) (*Supplementary file 3*). This response was also strongly attenuated in cells with reduced PSMD2 levels (*Figure 3C,D*, and *Figure 3—figure supplement 1C*). Overall, selectively compromising 19S subunit expression broadly reduces the diverse transcriptional responses that normally ensue when flux through the proteasome is reduced.

## Compromising 19S function primes a cell cycle response to bortezomib

To further characterize the transcriptional effects of 19S subunit reduction, we performed a cluster analysis on the genes displaying the highest differential expression in our RNA-seq experiment (*Figure 3E*). This analysis confirmed that PSMD2 reduction strongly blunted the bortezomib-mediated induction of stress response genes (FDR q-values 1.2 e$^{-5}$ to 1.2 e$^{-13}$). It also revealed broad changes in genes involved in small molecule metabolism, which remain to be deciphered.

One group of genes highlighted by this analysis revealed a connection between the suppression of the cell cycle and increased resistance to bortezomib. In cells with reduced levels of PSMD2, bortezomib treatment strongly repressed genes involved in DNA replication (FDR q-value = 1.4 e$^{-32}$) and cell cycle control (FDR q-value = 1.8 e$^{-90}$). These genes include replication factors, polymerases, cyclins, and cyclin-dependent kinases. This accentuated anti-proliferation response suggests that cells with reduced 19S subunits are primed to enter a protected, quiescent-like state when flux through the proteasome is compromised.

## Transiently reducing 19S subunits mirrors the effects of stable 19S subunit reduction

To model the effects of transient reduction of 19S subunits, we developed a cell line in which a PSMD2-targeting shRNA is transiently expressed from a doxycycline-regulated promoter (*Figure 4*). The effects of transient reduction of PSMD2 mirrored the effects of stable PSMD2 reduction. Most notably, it significantly increased resistance to both bortezomib (*Figure 4B*, *Figure 4—figure supplement 1A*) and MG132 (*Figure 4—figure supplement 1B*). Again, this resistance was selective, and not accompanied by increased resistance to other small molecule stressors (*Figure 4—figure supplement 1C–F*). In the absence of bortezomib, transient 19S reduction did not activate NRF1 or HSF1 (*Figure 4C*) and reduced activation of NRF1 and HSF1 following bortezomib exposure (*Figure 4C*).

## 19S subunit reduction counteracts the effects of bortezomib on proteasome degradation capacity and protein translation

We used native gels and glycerol gradient fractionation to monitor the levels and activity of proteasome complexes following transient reduction of PSMD2 (*Figure 4D,E*, *Figure 4—figure supplement 1G,H*).

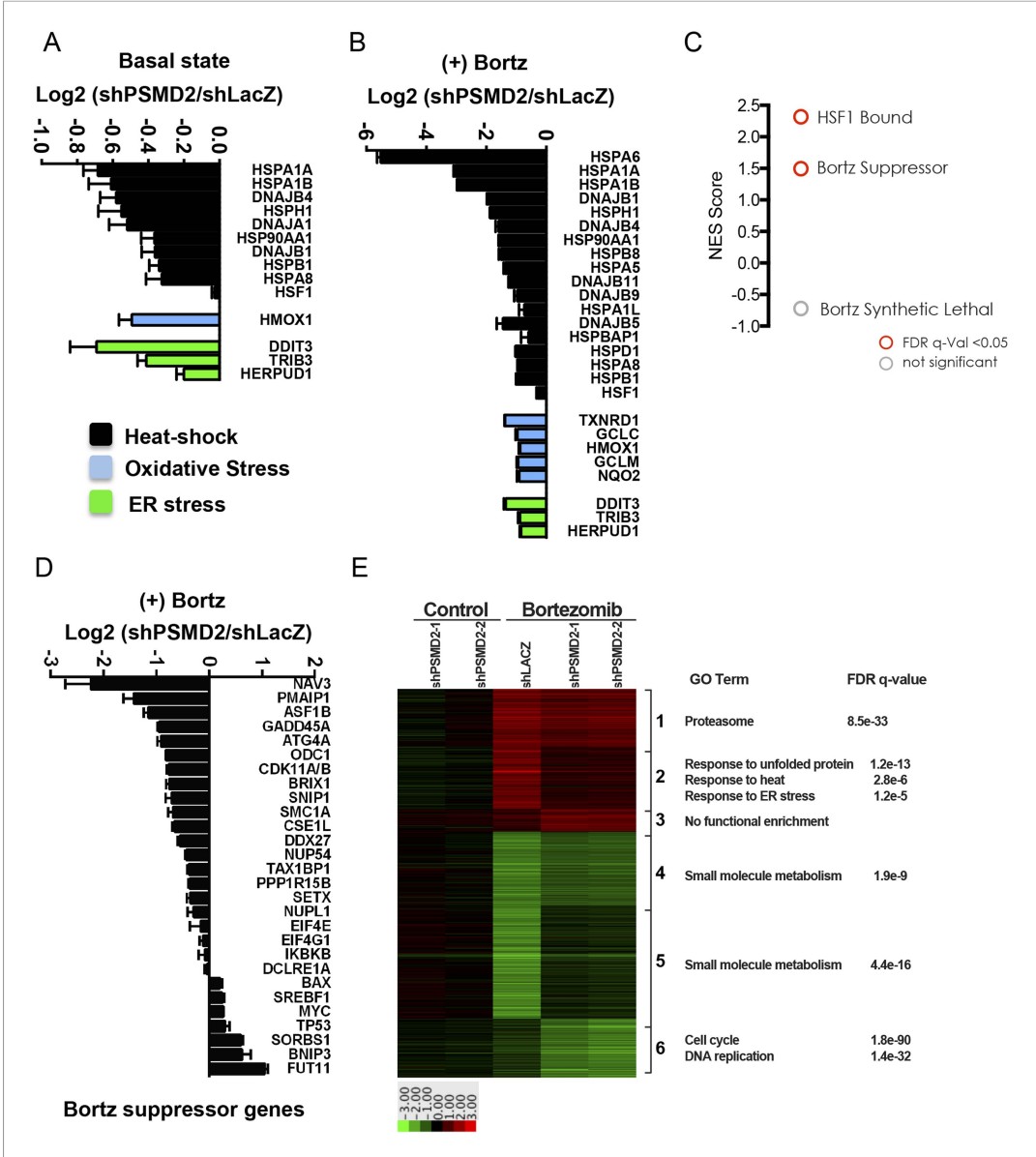

**Figure 3**. Inhibition of bortezomib-mediated transcriptional responses in PSMD2 knockdown cells. RNA-seq gene expression profiling was conducted on HepG2 cells that harbor two different shRNAs targeting PSMD2 (PSMD2-1, PSMD2-2; same cells used in *Figure 2F*) and on control cells (shLacZ). The effects of reducing PSMD2 levels on gene expression were highly correlated in both basal conditions (Pearson's r = 0.99) and following bortezomib treatment (Pearson's r = 0.94). (**A, B**) Heat-shock- (black), oxidative stress- (blue), and ER stress- (green) related gene expression were all lower in the PSMD2 knockdown cells vs control cells under both basal conditions (**A**) and upon introduction of 12 nM bortezomib for 24 hr (**B**). (**C**) Gene set enrichment analysis of genes upregulated in control but not in PSMD2 shRNA cells following bortezomib treatment. Enrichment was calculated for the indicated gene sets and is presented as a normalized enrichment score (NES). Statistically significant enrichment (false discovery rate [FDR] q-value < 0.05) is shown in red; non-significant enrichment is shown in gray. (**D**) Expression levels of genes previously characterized as suppressors of bortezomib-induced toxicity (*Chen et al., 2010*) are down-regulated in the PSMD2 knockdown cells following the addition of bortezomib. (**E**) Heat map depicting fold change in mRNA levels of genes differentially expressed in cells harboring control shRNA or PSMD2 shRNAs in the presence or absence of 12 nM bortezomib. Gene ontology enrichment is shown to the right of the panel.

The following figure supplement is available for figure 3:

**Figure supplement 1**. (**A**) Gene set enrichment analysis using the set of genes that are bound by HSF1 in MCF7 cancer cells under 37° basal conditions (*Mendillo et al., 2012*) was performed on genes negatively regulated in PSMD2 knockdown cells (siPSMD2) vs control cells (LacZ).

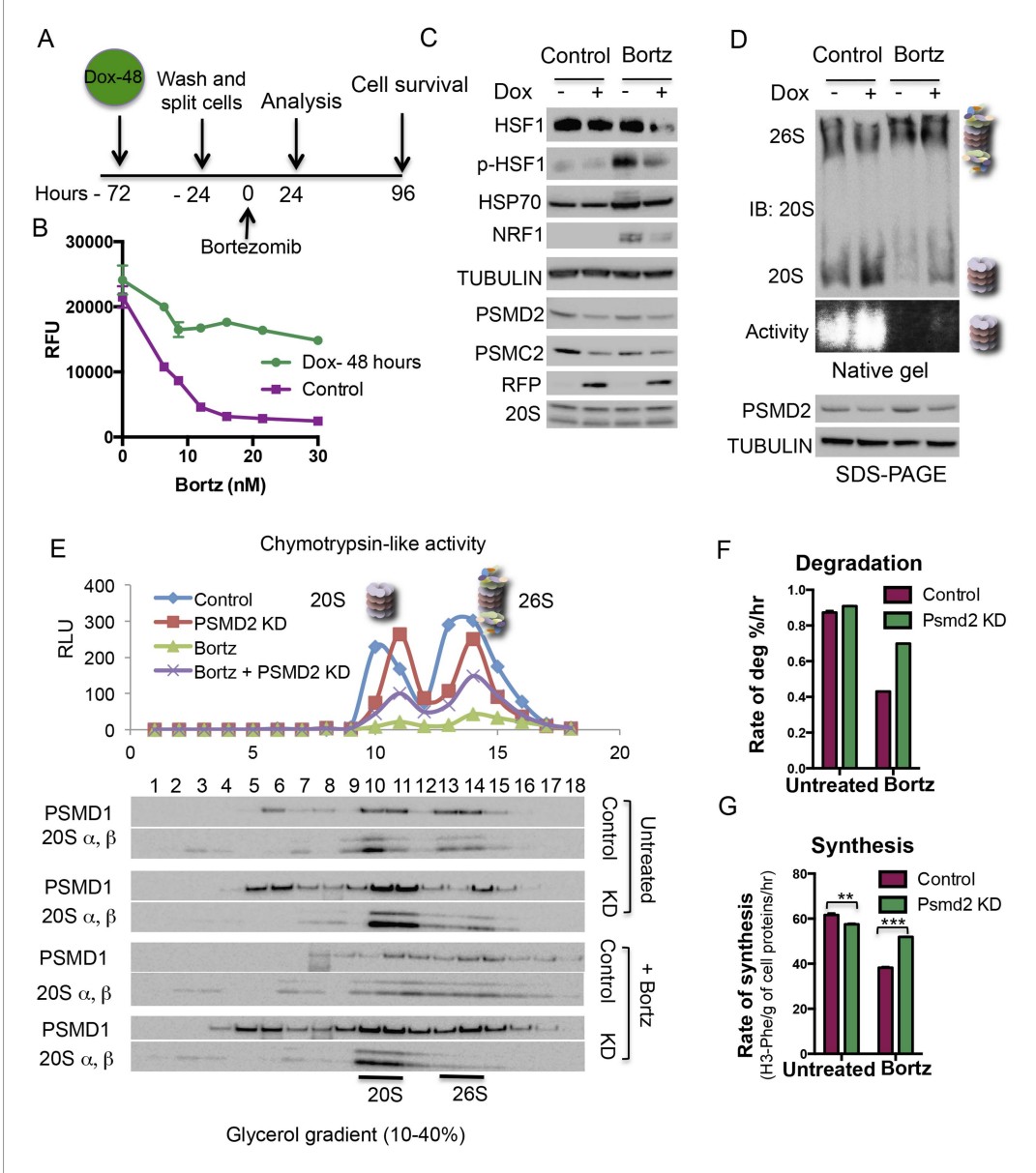

**Figure 4**. Transient induction of PSMD2 shRNA is sufficient to promote resistance to proteasome inhibition. (**A**) Schematic representation of the experimental design. (**B**) T47D cells harboring a doxycycline-inducible PSMD2 shRNA were grown in the presence or absence of 1 µg/ml doxycycline for 48 hr. Cells were then collected, washed, and plated in the absence of doxycycline for 24 hr prior to exposure to increasing concentrations of bortezomib. Relative cell numbers were measured 3 days later. (**C**) Protein content analysis by immunoblot for the indicated proteins on lysates from control or cells pre-treated for 48 hr with doxycycline (Dox), followed by a recovery of 24 hr and then incubation with 10 nM bortezomib for an additional 24 hr. (**D**) Native gel analysis of proteasome complexes in cells pre-treated as in (**C**). The proteasome complex levels and activity of the 20S proteasome were assessed by native gel electrophoresis. Loading controls were analyzed by immunoblot for PSMD2 and tubulin following SDS-PAGE. (**E**) Glycerol gradient fractionation (10–40%) was conducted on cells pre-treated for 48 hr with doxycycline (Dox), followed by a recovery of 24 hr and then incubation with 15 nM bortezomib for an additional 24 hr. Proteasome activity in the fractions collected was assessed with proteasome-Glo and proteasome content was analyzed by immunoblotting with PSMD1 and 20S subunits specific antibodies. (**F**) The rate of degradation was analyzed in cells with reduced levels of PSMD2 (green bars) vs control (red bars) in the presence or absence of 10 nM bortezomib (treatment for 20 hr) by monitoring the release of H3-Phe in pre-labeled cells. (**G**) Rate of total protein synthesis was determined in cells with reduced levels of PSMD2 (green bars) vs control (red bars) in the presence or

*Figure 4. continued on next page*

*Figure 4. Continued*

absence of 10 nM bortezomib (treatment for 20 hr) by measuring the rate of incorporation of 3H-phenylalanine for 1 hr. The p-values were obtained by conducting a two-tailed unpaired t-test. **p < 0.01, ***p < 0.001.

The following figure supplement is available for figure 4:

**Figure supplement 1**. (**A–F**) PSMD2 shRNA was induced for 48 hr with 1 µg/ml doxycycline.

Transient reduction of this 19S subunit increased the ratio of 20S/26S proteasomes and total levels of 20S proteasome activity (*Figure 4D,E*) without increasing total levels of 20S subunits (*Figure 4C*, *Figure 4—figure supplement 1G,H*). Similar to the results we obtained in stable knockdown cells (*Figure 2E*), transient 19S subunit reduction preserved a fraction of the 20S proteasome complexes following bortezomib treatment and a corresponding portion of their activity remained (*Figure 4D,E*, *Figure 4—figure supplement 1G,H*).

Next, we examined the impact of reducing PSMD2 levels on protein degradation and protein synthesis in cells labeled with tritiated-phenylalanine for 24 hr. We then separately measured the rates at which labeled proteins were degraded through the proteasome and through the lysosome. In the absence of bortezomib, transiently reducing PSMD2 lowered rates of proteolysis by the lysosome (*Figure 4—figure supplement 1I*), but it did not affect protein degradation by the proteasome (*Figure 4F*). This suggests that the 26S proteasome is normally present in excess.

To measure rates of protein synthesis, cells were pulse labeled with tritiated-phenylalanine for 1 hr. Reducing PSMD2 levels resulted in a significant and reproducible 7% decrease in the rate of protein synthesis (*Figure 4G*). Thus, even though PSMD2 knockdown did not reduce protein degradation capacity, it did trigger a reduction in protein translation (*Figure 4G*), a change that may contribute to lowering basal levels of proteotoxic stress.

Next, we measured protein degradation and synthesis rates after a 20-hr treatment with bortezomib. The rate of protein degradation sharply decreased in control cells (*Figure 4F*). Rates of proteolysis by the lysosome remained unchanged (*Figure 4—figure supplement 1I*). Transiently reducing PSMD2 strongly counteracted the inhibition of proteasome degradation (*Figure 4F*). Reducing the levels of 19S subunits partially preserved proteasome complex levels following the inhibition of flux through the proteasome by bortezomib. Correspondingly, it partially preserved 20S activity and the cells overall ability to degrade proteins (*Figures 2, 4D,E*, *Figure 4—figure supplement 1G,H*). Following bortezomib treatment, the rate of protein synthesis also sharply decreased in control cells (*Figure 4G*). This drop reflects the global repression of protein synthesis that normally follows strong proteotoxic stress (*Holcik and Sonenberg, 2005*; *Shalgi et al., 2013*). Transiently reducing PSMD2 protein levels strongly counteracted the bortezomib-mediated suppression of protein synthesis (*Figure 4G*). Thus, reducing PSMD2 levels counteracts the effects of bortezomib on both protein degradation and protein synthesis.

## Lower 19S subunit expression levels correlate with increased resistance to proteasome inhibitors across a broad spectrum of cancer cells

Human cancer cell lines have a broad range of sensitivities to proteasome inhibition. We asked if this might correlate with changes in 19S subunit expression. We analyzed the Genomics of Drug Sensitivity in Cancer (GDSC) database, a public resource of transcriptional data and drug responsiveness collected from a spectrum of human cancer cell lines with diverse tissue origins and diverse oncogenic lesions (*Garnett et al., 2012*). We ranked the 310 cell lines in the data set by their half maximal inhibitory concentration (IC50) to either MG132 or to bortezomib (highest to lowest). The cells comprising the top 10% were defined as the 'resistant' group and those in the bottom 10% were defined as the 'sensitive' group.

From the 31 cell lines in each group, we averaged the expression levels of all of the 20S subunits (*PSMA* and *PSMB* mRNA) and the expression levels of all of the 19S subunits (*PSMC* and *PSMD* mRNA). We found no significant difference in the average expression of 20S subunits between the two groups (*Figure 5A,B* left panels). However, cells that were the most resistant to either MG132 or to bortezomib had significantly lower levels of 19S transcripts (*PSMC* and *PSMD* mRNA) than cells that

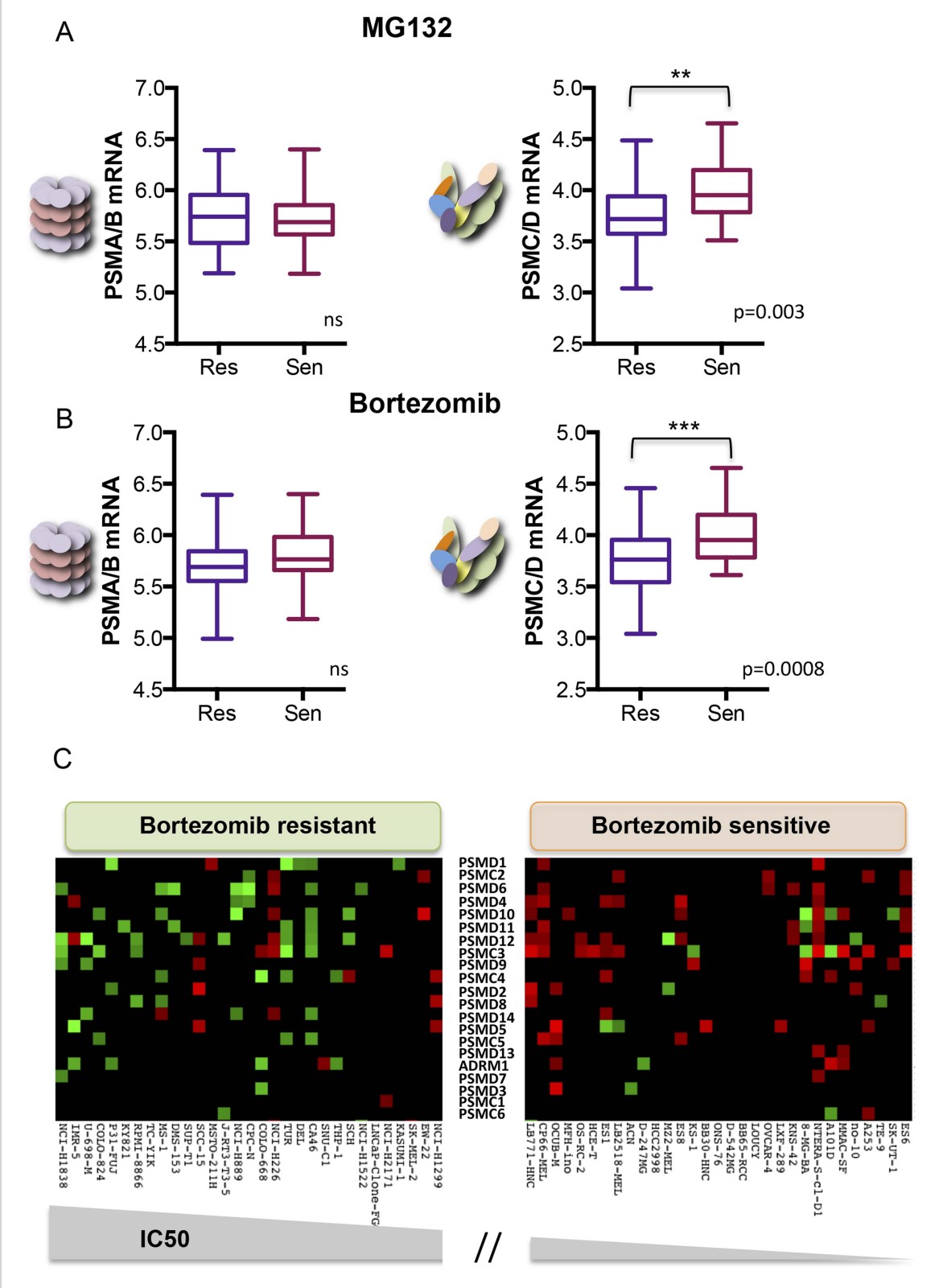

**Figure 5**. Reduced expression of 19S subunits correlates with resistance to proteasome inhibitors. (**A**, **B**) Analysis of expression data from 315 cell lines in the Genomics of Drug Sensitivity in Cancer (GDSC) database (*Garnett et al., 2012*). The levels of 20S proteasome subunit (PSMAs and PSMBs) gene expression (**A** and **B** left panels) and 19S subunit (PSMCs and PSMDs) gene expression (**A** and **B** right panels) were analyzed in the cell lines that are the

*Figure 5. continued on next page*

Figure 5. Continued

10% most sensitive or the 10% most resistant to either MG132 (**A**) or bortezomib (**B**). (**C**) The relative expression level of each 19S complex subunit was analyzed in the bortezomib resistant and sensitive groups. Expression levels with deviation of more than twofold from the average were color-coded (red-up, green-down). The p-values were obtained by conducting a two-tailed unpaired t-test. **p < 0.01, ***p < 0.001.

The following figure supplement is available for figure 5:

**Figure supplement 1**. The relative expression level of each 19S complex subunit was analyzed in the MG132 resistant and sensitive groups.

were sensitive (*Figure 5A,B* right panels; p-value = 0.003 for MG132; p-value = 0.0008 for bortezomib). This observation is striking as the expression levels of all proteasome subunits, both 20S and 19S, are regulated by similar mechanisms and are normally highly correlated (*Jansen et al., 2002*; *Radhakrishnan et al., 2010*, *2014*; *Sha and Goldberg, 2014*).

We next assessed the expression of the individual 19S regulatory complex subunits in each of the resistant and sensitive cell lines. A heat map of genes with significantly altered expression (>twofold deviation from average) revealed that bortezomib-sensitive cells commonly showed increased expression of many different 19S subunits (*Figure 5C*, right panel-red). Resistant cells generally had at least a twofold reduction in expression of one or more 19S subunits (*Figure 5C*, left panel-green). This was also true in the case of MG132 (*Figure 5—figure supplement 1*). Thus, alterations in 19S subunit expression commonly occur in the evolution of cancer cells.

## Transiently reducing a 19S subunit confers a competitive survival advantage in the face of protein flux inhibition

Human cancers are increasingly viewed as complex ecosystems comprised of cells harboring enormous genetic, functional and phenotypic heterogeneity (*Meacham and Morrison, 2013*). We asked if heterogeneity arising from 19S subunit expression can alter population dynamics and confer a fitness advantage in the face of exposure to proteasome inhibitors. To do so, we investigated the effects of transiently reducing PSMD2 expression in only a subpopulation of cells.

We created two cell lines—one line that expresses red fluorescent protein (turboRFP) and the doxycycline-inducible PSMD2-targeting shRNA and another line that expresses green fluorescent protein (GFP) and a doxycycline-inducible control shRNA (*Figure 6A*). First, we induced shRNA expression with doxycycline for 48 hr. After recovery, we mixed shPSMD2-RFP and shControl-GFP cells at different ratios (1:1, 1:2, 1:5, or 1:10), adding the cells with reduced PSMD2 as the minority subpopulation. 24 hr after plating, we treated these mixed populations of cells for 48 hr with increasing concentrations of bortezomib (5, 7.5, or 10 nM). We allowed the cells to recover in the absence of bortezomib and then we quantified the red and green cells by fluorescence-activated cell sorting (FACS) analysis (*Figure 6A*) and captured representative images by fluorescence microscopy (*Figure 6B*).

In the absence of proteasome inhibitors, the initial plating ratios of these cells were maintained for 6 days (1:1, 1:2, 1:5, and 1:10) (*Figure 6A*). In contrast, even low concentrations of bortezomib (5 nM) substantially shifted the populations of surviving cells and higher concentrations (7.5 nM and 10 nM) elicited even more substantial shifts (*Figure 6A,B*). In the presence of proteasome inhibitors, cells with modestly reduced levels of PSMD2 had a strong competitive advantage.

## The protective effect of 19S subunit reduction is conserved in yeast

Finally, because the essential role of the proteasome in maintaining protein homeostasis is conserved across all eukaryotes (*Hilt and Wolf, 1995*), we asked whether reducing expression of 19S subunits confers resistance to proteasome inhibitors in an evolutionary distant organism—the yeast *Saccharomyces cerevisiae*. As the 19S regulatory complex components are essential for viability in yeast, we utilized a library of hypomorphic (DAmP) alleles for essential yeast genes. In this library, the expression of individual mRNA species is reduced from two to 10-fold by replacing the mRNA's 3′ untranslated region (*Breslow et al., 2008*).

DAmP-strains were available for 12 genes comprising the 19S regulatory complex. These strains showed no significant growth-impairment under basal conditions (*Figure 7—figure supplement 1*). However, five of these twelve strains had significantly increased resistance to proteasome inhibition

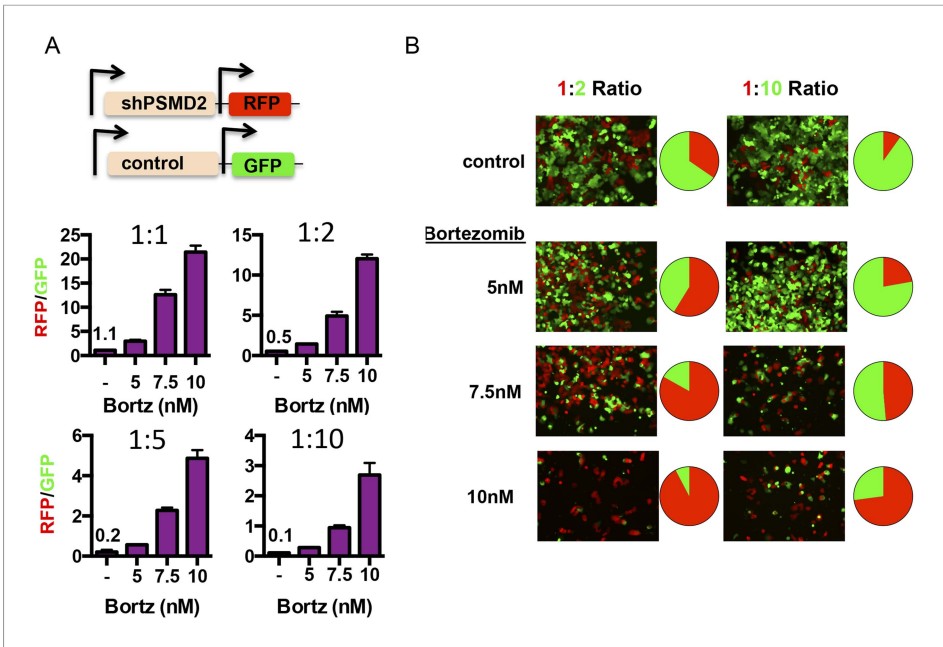

**Figure 6**. Transient 19S subunit reduction confers a competitive survival advantage in the presence proteasome inhibitors. (**A**, **B**) T47D cells that harbor a doxycycline-inducible control shRNA (GFP) or a doxycycline-inducible PSMD2 shRNA (TurboRFP) were incubated with doxycycline for 48 hr. Cells were collected, counted, and plated at the indicated ratios of TurboRFP-expressing PSMD2 shRNAs/GFP expression control shRNAs (1:1, 1:2, 1:5, and 1:10). 24 hr later, bortezomib was added at the specified concentrations and incubation continued for 48 hr. Cells were allowed to recover in the absence of bortezomib for another 48 hr and then visualized by microscopy (**B**) or analyzed by FACS after 6 days of recovery (**A**, and pie charts in **B**). The green and red images were overlaid using ImageJ. DOI: 10.7554/eLife.08467.013

by MG132 (*Figure 7*). Most notable were Rpn5 and Rpt6, the yeast orthologs of PSMD12 and PSMC5—the two most significantly enriched genes in our MG132 screen in human cells (*Figure 1C*).

## Discussion

We have identified a highly conserved mechanism that enables organisms as diverse as yeast and humans—separated in evolution by over 1 billion years—to withstand inhibition of protein flux through the proteasome. Very surprisingly, when the proteasome is inhibited to toxic levels, suppressing individual components of the 19S regulatory complex increases cell survival. While strong reduction of any of these subunits is not tolerated, modest reduction is protective.

In this partially protective state, 26S proteasomes decrease and the levels and activity of 20S proteasomes sharply increase. In the absence of proteasome inhibitors, the change in 20/26S proteasome complex ratios does not reduce protein degradation, polyubiquitinated substrates are not elevated, and hallmark stress responses are not activated. Moreover, at concentrations of proteasome inhibitors that normally unleash these responses, they are suppressed.

A number of mechanisms could potentially mediate this protective effect. Our data support a significant role for shifts in the ratio of 20S/26S proteasome complexes. Normally, bortezomib treatment suppresses proteasome catalytic activity. We find that bortezomib treatment also leads to a sharp decrease in the levels of proteasome complexes. Remarkably, the protection against bortezomib that is conferred by reducing 19S subunits correlates with a preservation of both the activity of the proteasome and of the levels of the proteasome complexes themselves. Other mechanisms that may contribute include the induction of autophagy, the induction of alternative proteasome regulatory components, or the disruption of normal gating and drug–target interaction affinities. These possibilities require further investigation. However, our general findings on the surprising ability of 19S subunit reductions to increase resistance to proteasome inhibitors are complementary to those of D Acosta-Alvear, P Walter, JS Weissman, and M Kampmann (Personal Communication) who arrived at this conclusion through very different approaches.

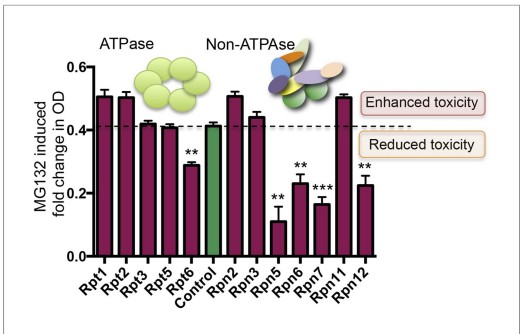

**Figure 7**. Reducing the levels of 19S subunits is an evolutionarily conserved mechanism to acquire resistance to proteasome inhibition. Proteasome subunit DAmP strains and the BY4741 control strain were grown in the presence or absence of 50 μM MG132 for 48 hr. The relative change in OD induced by MG132 is plotted. Five proteasome subunit DAmP strains exhibited significantly reduced toxicity in the presence of MG132. **p < 0.01, ***p < 0.001.

The following figure supplement is available for figure 7:

**Figure supplement 1**. Proteasome subunit DAmP strains and the BY4741 control strain were grown in YPD media and OD600 was measured after 48 hr.

While changes in the ratio of 20S/26S proteasomes have not previously been shown to protect cells from inhibition of flux through the proteasome, it is well documented that they occur. Indeed, a broad range of genetic, metabolic, and environmental factors can elicit such changes. In human stem cells, manipulation of just a single subunit of the 19S regulatory complex modified the 20S/26S proteasome ratio (*Vilchez et al., 2012*). In cancer cells, many chromosomal regions are recurrently lost, and these often harbor genes that encode 19S subunits (*Nijhawan et al., 2012*; *Davoli et al., 2013*). Indeed, mining data from a survey of 310 cancer cell lines, we find that those that have increased resistance to proteasome inhibitors have strongly reduced expression of at least one but often of several genes encoding 19S subunits.

The metabolic and environmental factors that can elicit reversible shifts in the 20S/26S proteasome ratio are many and varied. For example, nutrient depravation in yeast (*Bajorek et al., 2003*), the activation of glutamate receptor signaling in neurons (*Tai et al., 2010*), mitochondrial dysfunction in yeast and mammalian cells (*Livnat-Levanon et al., 2014*), and various states of increased oxidative stress (*Wang et al., 2010*; *Livnat-Levanon et al., 2014*) can all increase the levels of 20S proteasome complexes. The ratio of proteasomes can also be regulated by cellular levels of NADH, a co-enzyme that directly binds 19S subunits and influences 26S proteasome stability (*Tsvetkov et al., 2014*). Further, the ratio is shifted by post-translational modifications mediated by $NAD^+$ ADP-ribosyltransferases (*Ullrich et al., 1999*; *Cho-Park and Steller, 2013*).

The modest reductions in 19S subunits in our experiments did not reduce the overall rates of protein degradation and did not activate proteotoxic stress responses suggesting that these cells normally have a buffer, or excess, of 26S proteasomes. Such a buffer has been recently described in neurons (*Asano et al., 2015*), and if true more broadly, might generally allow cells to tolerate reductions in 19S subunit expression without altering basal rates of proteolysis by the proteasome. In our hands, 19S subunit reduction was accompanied by accumulation of active 20S proteasome complexes. These complexes are highly effective in degrading oxidized (*Reinheckel et al., 1998*; *Grune et al., 2003*) and intrinsically disordered proteins (*Baugh et al., 2009*; *Tsvetkov et al., 2009a*; *Wiggins et al., 2011*; *Ben-Nissan and Sharon, 2014*) in an ubiquitin-independent manner. Our results suggest that cells with expanded 20S capacity might be even more broadly positioned to cope with toxic products that accumulate following inhibition of the proteasome.

The increase in 20S proteasomes may also have other, pleiotropic effects that contribute to the protective state. First, these complexes mediate the endoproteolytic cleavage of translation initiating factors eIF4G1, eIF4F, and eIF3a (*Baugh and Pilipenko, 2004*), which could directly contribute to the inhibition of protein synthesis that occurs following 19S subunit reduction. Second, 20S proteasomes preferentially degrade newly synthesized substrates (*Tsvetkov et al., 2009b*; *Adler et al., 2010*). In addition, 20S proteasome complexes degrade numerous intrinsically disordered proteins involved in cell cycle regulation, cell cycle control, and oncogenesis (*Asher et al., 2006*; *Jariel-Encontre et al., 2008*; *Ben-Nissan and Sharon, 2014*). The degradation of such 20S substrates could underlie the robust anti-proliferation response that follows bortezomib treatment of cells with reduced 19S subunits. Such a shift into a quiescent-like state likely triggers adaptive cytoprotection. This is reminiscent of yeast that transition into stationary phase when nutrients are exhausted. In this setting, there is also a reversible reduction in protein translation (*Fuge et al., 1994*) and levels of 26S proteasomes sharply decrease in favor of active 20S proteasomes, which are essential for viability

during prolonged periods of nutrient depravation (*Bajorek et al., 2003*). Thus, the protective mechanism that is generated upon 20S formation is likely conserved from yeast to human and may be part of the natural transitions used to established stress-resistant quiescent states.

In our experiments, this mechanism for increasing resistance was revealed by the use of highly controllable chemical compounds. However, in nature, mechanisms for rebalancing the 20S/26S proteasome ratio most likely emerged to help cells contend with perturbations that cause protein misfolding. In fact, intracellular and environmental insults that generate large protein aggregates are known to impair the proteolytic function of the proteasome (*Deriziotis et al., 2011*; *Ayyadevara et al., 2015*). Such mechanisms may have also helped organisms withstand naturally occurring 20S proteasome inhibitors that are elaborated by microorganisms cohabiting their niches (*Schneekloth and Crews, 2011*). Such selective pressures could have shaped the evolution of this ancient survival mechanism, one that emerged long before the advent of the use of proteasome inhibitors as anti-cancer therapeutics. In agreement with our results indicating that yeast cells are protected from proteasome inhibitors by reducing 19S subunits, Breslow et al. found that reducing 19S subunits can rescue yeast strains that are growth inhibited by reductions in 20S subunits (*Breslow et al., 2008*).

Suppressing the expression of many different 19S subunits provided resistance to proteasome inhibitors and there are many potential routes for suppressing their expression (e.g., genetic, metabolic, epigenetic, environmental). This raises the intriguing possibility that large populations of cells might harbor functional heterogeneity for surviving altered flux through the proteasome. At one extreme, some cells might be highly proliferative yet highly sensitive to proteasome inhibition, while at the other extreme, some cells could be slowly proliferative yet highly resistant to proteasome inhibition. Because of their slower proliferation capacity, the latter would have generally reduced relative fitness, analogous to the small populations of drug-tolerant 'persister' cells that reside within tumor populations (*Sharma et al., 2010*; *Glickman and Sawyers, 2012*; *Knoechel et al., 2014*). Developing a strategy to address this state of resistance could have significant therapeutic value.

## Materials and methods

### Experimental procedures

#### Screening

The construction of gene-trap viral vectors, generation of mutagenized KBM7 libraries, mapping of insertion sites, and screening approach were performed as described previously (*Carette et al., 2009*, *2011a*, *2011b*). We performed pilot experiments to determine the concentrations of MG132 and bortezomib that would allow the emergence of resistant clones from a pilot collection of mutagenized KBM-7 cells following a 10-day incubation. 700 nM MG132 and 18 nM bortezomib were found to be optimal concentrations. 100 million mutagenized cells were exposed to 700 nM MG132 and 18 nM bortezomib and resistant clones were expanded and pooled. Genomic DNA was isolated, and a PCR-based approach was followed to amplify the retroviral insertion sites followed by Illumina sequencing. Mutations that were predicted to be disruptive in genes were counted per gene and compared to mutation frequencies in the same gene in a non-selected cell population. Genes significantly enriched for mutations in the selected cell population were identified. Deep-sequencing data have been deposited in the NCBI Sequence Read Archive under accession number: PRJNA281714.

#### Cell culture methods

HEK293T and HepG2 were cultured in Dulbecco's modified Eagle's medium supplemented with 10% fetal bovine serum; H838, H1792, T47D were cultured in RPMI-1640 medium supplemented with 10% fetal bovine serum.

#### Small hairpin knockdown of proteasome subunits and controls

For the analysis of multiple proteasome subunit knockdown, 80 shRNA targeting 20 proteasome subunits (4 different shRNAs) and 13 control shRNAs (*Supplementary file 2*) in pLKO lentiviral vectors from the RNAi consortium shRNA library were utilized. Cells were plated in 96 wells with the volume of 100 μl media at the concentration of 2500 cells/well. 24 hr after plating, media was discarded and 50 μl with 7.5–10 μg/μl of polybrene and 7 μl of purified virus was added. After incubation for 24 hr, the media was discarded and 200 μl of fresh media with 1 μg/ml puromycin was added. Bortezomib was added where specified. pLKO lentiviral vectors from the Broad Institute RNAi consortium

(Cambridge, MA) shRNA library targeting the PSMC5 and PSMD2 were further used to create HepG2 and T47D stably overexpressing these shRNAs by selection with puromycin 1 µg/ml for 1 week.

For the generation of the T47D Tet-inducible PSMD2 knockdown cell line, the TRIPZ vector with an inducible shRNA targeting PSMD2 was purchased from Dharmacon (clone V3THS_403760). It was introduced to the T47D cells according to manufactures protocol and cells were selected with puromycin 1 µg/ml for 1 week. The cells were exposed to doxycycline for 24 hr and cells were FACS sorted for the top 10% of most RFP expressing cells (highest expression of shRNA). The cells were further cultured in the absence of doxycycline and PSMD2 knockdown was induced as specified in the text. The TRIPZ control GFP vector was created by removing the turboRFP from the TRIPZ control vector by digestion with AgeI and ClaI and replacing it with GFP amplified with primers flanking with AgeI and ClaI restriction sites. Primers: 5′ Primer for GFP- AAAAAACCGGTCGCCACCatggtgag caagggcgagga, 3′ Primer for GFP- TTTTTATCGATTActtgtacagctcgtccatgccga.

## Generation of mutant PSMD12 and PSMC2 ES cells

PSMD12 and PSMC2 AS clones were infected with a retro virus carrying CRE IRES fusion gene under the control of CMV (cytomegalovirus) promoter. 72 hr after the infection, Cre + cells were sorted by FACS and expanded in ES cell medium. After 3–4 weeks, individual subclones were picked, separately plated, further expanded, and finally analyzed for successful inversion event and PSMD12/PSMC2 gene expression.

Primer sequences for genotype analysis (Cre inversion) are (standard PCR conditions):

F1: TCGACCTCGAGTACCACCACACT
F2: AAACGACGGGATCCGCCATGTCA
R1: TATCCAGCCCTCACTCCTTCTCT

Primer sequences for gene expression analysis are (standard RT PCR conditions):

PSMD12 F: CTGTGGATGAGTCAGAGGCT
PSMD12 R: TTGGCTATGAGGTGTGTCGT
PSMC2 F: ACAGCCATTACAGGTGGCAA
PSMC2 R: GTCCACACCGACTCTCATCC

## Visualization and FACS analysis of GFP/RFP levels

Cells were trypsinized in 100 µl Accumax solution and further diluted into 100 µl PBSx1. The number of cells with red or GFPs was measured by MACSQuant VYB according to manufactures protocol. Images of RFP and GFP were created by overlaying images using Fiji software.

## Protein level expression analyses

For the analysis of protein expression, cells were lysed in HENG buffer (50 mM Hepes-KOH pH 7.9, 150 mM NaCl, 2 mM EDTA pH 8.0, 20 mM sodium molybdate, 0.5% Triton X-100, 5% glycerol, 0.2 mM PMSF, 1 mM NaF, and protease inhibitor cocktail [Roche Diagnostics, Cat# 11836153001], Indianapolis, IN, United States). Protein concentration was determined by the BCA assay (Thermo Fisher Scientific 23227, Asheville, NC, United States) and proteins were resolved on SDS-PAGE for immunoblot analysis. The antibodies used are specified in the *Supplementary file 2*.

## Compounds used

The following compounds were used. MG132 (EMD Millipore, Billerica, MA, United States), Bortezomib (LC Laboratories # B-1408, Woburn, MA, United States), Cyclohexamide (Enzo life sciences, Farmingdale, NY, United States), withaferin A, tunicamycin (Sigma St. Louis, MO, United States).

## Cell viability assay

Relative cell growth and survival were measured in 96-well microplate format in the shRNA experiments and in 384-well format in the drug toxicity assays, by using the fluorescent detection of resazurin dye reduction as an endpoint (544-nm excitation and 590-nm emission). 2500 cells in 96-well format or 1000 cells in 384-well format were plated 24 hr before compound exposure (for 72 hr or indicated time). Each analysis was performed at least with three replicas.

## Gene expression analysis

RNA was extracted from triplicate samples and RNA libraries were prepared for sequencing by NEBNext Ultra RNA Library Prep Kit for Illumina (New England BioLabs, Ipswich, MA), including the removal of large and small RNA, synthesis of cDNA, and construction of cDNA libraries. Libraries were barcoded using NEBNext Multiplex Oligos for Illumina (NEB). Libraries were sequenced using Illumina HiSeq 2500,

with paired-end 100bp reads. Paired-end reads were aligned to UCSC human transcriptome 19 (hg19) using TopHat (Bowtie v2.0.9). Alignment quality and read distribute was assessed via SAMtools (v0.1.19). Transcript assembly was conducted using cufflinks (v2.2.1). Normalized expression data were generated from aligned BAM files using cuffnorm and cuffdiff. Transcripts with zero values for FPKM (fragments per kilobase of transcript per million mapped) across all samples were removed. The mean for the triplicate technical replicates was created, and after adding 1 pseudocount count, was log2-normalized. The resulting gene expression matrix is presented in *Supplementary file 3*. RNA-sequencing data have been deposited in the NCBI Sequence Read Archive under accession number: PRJNA281613.

Genes differentially expressed upon PSMD2 knockdown in either the presence or absence of bortezomib treatment were determined as follows (*Figure 3E*): for each gene in the matrix described above, the values were normalized to the average expression in the LacZ control cells. Genes for which the absolute value of any condition vs control was greater than 1 and whose expression was significantly different between any condition (<0.05 p-value in a student's t test) vs control were included. These differentially expressed genes were clustered by k-means clustering.

Selective gene set enrichment analysis (*Figure 3C* and *Figure 3—figure supplement 1*) was conducted by using GSEA v2.2.0 software. Genes without detectable levels of expression across all samples within the individual analyses were excluded. The metric used for ranking genes was the difference of classes. The gene-sets 'HSF1 bound' and 'heat-shock up' were derived from GEO (GSE45851) (*Mendillo et al., 2012*). The gene-sets 'bortezomib suppressor' and 'bortezomib synthetic lethal' were obtained from Table 1 in (*Chen et al., 2010*). Gene ontology enrichment (*Figure 3E*, *Supplementary file 3*) was calculated using GOrilla software (*Eden et al., 2009*).

## Genomics of drug sensitivity in cancer data analysis

The IC50 values for bortezomib and MG132 across 315 cancer cell lines were obtained from http://www.cancerrxgene.org/ (*Garnett et al., 2012*; *Yang et al., 2012*). Gene expression data were obtained from the Oncomine Platform (*Supplementary file 4*). The average gene expression for the genes that comprise the 20S proteasome subunit (*PSMAs* and *PSMBs*) and the average gene expression for the genes that comprise the 19S subunit (*PSMCs* and *PSMDs*) was analyzed in the cell lines that are the 10% most sensitive or the 10% most resistant to either MG132 or bortezomib. The p-values were obtained by conducting a two-tailed unpaired t-test.

## Translation and degradation assays

For measuring overall rate of synthesis, cells were pretreated with 10 nM Bortezomib for 20 hr, then incorporation of $^3$H-phenylalanine was measured for 1 hr. The rate of synthesis was described as counts incorporated into cell proteins per hour and per μg of total cell proteins. When working with T47D cells, knockdown of PSMD2 was introduced for 3 days with doxycycline before bortezomib treatment. For measuring overall rates of protein degradation, pulse-labeling with $^3$H-phenylalanine for 24 hr was done before the bortezomib treatment as previously described (*Zhao et al., 2007*).

## Glycerol gradient fractionation

T47D cells harboring the doxycycline inducible PSMD2 KD were grown to confluency; cells were then grown in the presence or absence of 1 μg/ml of doxycycline for 48 hr (media changed every 24 hr). Cells were washed, collected, and lysed in buffer containing (50 mM Tris 7.5, 5% glycerol, 150 mM NaCl, 5 mM MgCl2, 1 mM ATP, 1 mM DTT, 0.1% Triton-x100). Protein concentration was assessed using the BCA method and 1 mg of protein cell extract was loaded onto the glycerol gradient. The gradients were subjected to ultra-centrifugation in SW-41 (100,000×*g*) for 16 hr. 18 fractions were collected and analyzed for proteasome activity by proteasome-Glo (according to manufacturer's protocol) and for proteasome subunit levels by immunoblot analysis.

## Yeast strains and MG132 sensitivity assay

MG132 sensitivity protocol was conducted as previously described (*Liu et al., 2007*). Yeast cells were grown overnight in media containing L-proline as nitrogen source instead of ammonium sulfate. The overnight cultures were diluted into OD 0.1 and were grown in the L-proline culture with 0.003% SDS with or without 50 μM MG132. OD was measured over the period of 48 hr.

## Native gel analysis of proteasomal complexes

Proteasomal samples were loaded on a non-denaturing 4% polyacrylamide gel using the protocol described previously. Gels were either overlaid with Suc-LLVY-AMC (50 μM) for assessment of

proteasomal activity or transferred to nitrocellulose membranes where immunoblotting specific for proteasomal subunits was conducted. Proteasomal activity was assessed by measuring the hydrolysis of Suc-LLVY-AMC by substrate overlay assays in native polyacrylamide gels with 50 mM Tris-HCl, pH 7.8, 5 mM MgC12, 1 mM DTT, 2 mM ATP, 50 µM Suc-LLVY-AMC peptide, and incubating the gels at 37°C for 30–60 min. Activity was visualized by transillumination by a UV light and photographed with BioRad ChemiDoc imaging system.

## Acknowledgements

We are indebted to Linda Clayton, Ruthie Scherz-Shouval, Can Kayatekin, Georgios Karras, Luke Whitesell, and members of the Lindquist and Goldberg labs for helpful comments and helpful discussions. We thank Tom DiCesare for graphical assistance. We thank Josh Francis (Broad Institute) for assistance with RNA-seq and Yuxiong Feng and members of the Gupta lab for reagents and help. We thank the WIBR-GTC for sequencing support and the Koch Center Koch Institute Flow Cytometry Core at MIT for FACS analysis. P.T was supported by EMBO fellowship ALTF 739-2011 and the Charles A King trust postdoctoral fellowship program. SS was supported by the Koch Institute Frontier Research Program through the Kathy and Curt Marble Cancer Research Fund and from the Alexander and Margaret Stewart Trust, NIH K08NS064168, the Valvano Foundation, and the Jared Branfman Sunflowers for Life Fund for Pediatric Brain and Spinal Cancer Research.

## Additional information

### Funding

| Funder | Grant reference | Author |
| --- | --- | --- |
| EMBO | ALTF 739-2011 | Peter Tsvetkov |
| National Institutes of Health (NIH) | K08NS064168 | Sandro Santagata |
| The Medical Foundation (TMF) | Charles A. King Trust | Peter Tsvetkov |

The funders had no role in study design, data collection and interpretation, or the decision to submit the work for publication.

### Author contributions

PT, SS, Conception and design, Acquisition of data, Analysis and interpretation of data, Drafting or revising the article; MLM, JZ, JEC, PHM, Acquisition of data, Analysis and interpretation of data, Drafting or revising the article; DC, MV, Acquisition of data, Analysis and interpretation of data, Contributed unpublished essential data or reagents; FRD, Acquisition of data, Drafting or revising the article, Contributed unpublished essential data or reagents; JMP, ALG, Analysis and interpretation of data, Drafting or revising the article, Contributed unpublished essential data or reagents; TRB, Conception and design, Acquisition of data, Analysis and interpretation of data, Contributed unpublished essential data or reagents; SL, Conception and design, Analysis and interpretation of data, Drafting or revising the article

## Additional files

### Supplementary files

• Supplementary file 1. KBM7 screening hits for MG132 and bortezomib, insertions and p-values.

• Supplementary file 2. Validation reagents: lentivirus clones, selected shRNAs, antibodies.

• Supplementary file 3. RNA-seq analysis.

• Supplementary file 4. Genomics of drug sensitivity in cancer analysis.

## Major datasets

The following datasets were generated:

| Author(s) | Year | Dataset title | Dataset ID and/or URL | Database, license, and accessibility information |
|---|---|---|---|---|
| Tsvetkov P, Mendillo ML, Zhao J, Carette JE, Merrill PH, Cikes D, Varadarajan M, Diemen FR, Penninger JM, Goldberg AL, Brummelkamp TR, Santagata S, Lindquist S | 2015 | Homo sapiens KBM-7 Whole genome sequencing | http://www.ncbi.nlm.nih.gov/Traces/sra/?study=SRP057499 | PRJNA281714. |
| Tsvetkov P, Mendillo ML, Zhao J, Carette JE, Merrill PH, Cikes D, Varadarajan M, Diemen FR, Penninger JM, Goldberg AL, Brummelkamp TR, Santagata S, Lindquist S | 2015 | Homo sapiens Raw sequence reads | http://www.ncbi.nlm.nih.gov/Traces/sra/?study=SRP057446 | PRJNA281613. |

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
