## [Decision Letter]

Thank you for submitting your work entitled “Compromising the 19S proteasome complex protects cells from reduced flux through the proteasome” for peer review at *eLife*. Your submission has been favorably evaluated by Randy Schekman (Senior Editor) and two reviewers, one of whom, Jeffery W Kelly, is a member of our Board of Reviewing Editors.

The reviewers have discussed the reviews with one another, and the Reviewing editor has drafted this decision to help you prepare a revised submission.

The Lindquist Laboratory exposed cells to pharmacologic proteasome inhibitors and performed genome-wide screens for mutations that allowed the cells to remain viable. Survival was increased by reduced expression of subunits comprising the 19S regulatory subunit of the proteasome, thereby increasing the ratio of 20S to 26S proteasomes. The central question both reviewers would like discussed in the paper is: how is it that a reduction of 19S would render the cells less sensitive to inhibition by bortezomib?

The main concern of the expert reviewer has to do with the hypothesis put forward by the authors to explain why cells with diminished 19S cap are more resistant to bortezomib and/or MG132. The chief argument appears to be that a subset of the 20S is refractory to inhibition in these cells. However, there are some problems with the experiments that show this. In the first experiment (Figure 2 panel F), there is indeed less 20S activity in WT cells treated with bortezomib compared with 19S-deficient cells. However, there is also much less proteasome. Where is the proteasome going in the WT control? Is this simply a consequence of cell death? But compare with panel 2B – cell number actually seems to be increasing over the first 24h.

This experiment would benefit from being repeated as a time-course, so that it is possible to evaluate proteasome activity at a time before the proteasome has largely disappeared from the WT cells. It would also be much better to do this experiment with epoxomicin or carfilzomib because BTZ can potentially dissociate during the native gel electrophoresis, given that the t1/2 has been reported to be in the range of 100 minutes. Finally, it would be desirable to see this effect quantified (e.g. with an LLVY-AMC assay or similar). Please address as many of these suggestions as possible when revisiting your paper for serious consideration for publication.

Reviewer #1:

This very well written paper entitled “Compromising the 19S proteasome complex protects cells from reduced flux through the proteasome” is competitive for space in *eLife* from this reviewer's perspective.

The Lindquist Laboratory exposed cells to pharmacologic proteasome inhibitors and performed genome-wide screens for mutations that allowed the cells to remain viable. Survival was increased by reduced expression of subunits comprising the 19S regulatory subunit of the proteasome, thereby increasing the ratio of 20S to 26S proteasomes. While this cellular alteration has fitness costs in the absence of stress (stable cell lines could not be isolated under basal conditions), this means of counteracting reduced flux through the proteasome under stressful conditions is conserved from yeast to humans. All insertions that reached a high level of statistical enrichment occur in genes encoding subunits of the proteasome 19S regulatory complex. These included both subunits that hydrolyze ATP (*PSMC2*, *PSMC3*, *PSMC4*, *PSMC5*, and *PSMC6*) subunits that are not ATPases (*PSMD2*, *PSMD6*, *PSMD7*, and *PSMD12*). No insertions were recovered in genes encoding subunits of the 20S catalytic component of the proteasome. The Bortezomib screen uncovered less loss-of-function hits, consistent with its superior selectivity relative to MG-132 that hits many targets in addition to the 20S proteasome. ShRNA targeting of the PSMD2 subunit of the 19S regulatory subunit rendered cells much more resistant to Bortezomib without activating the heat shock response or the NRF2 transcription factor-this treatment did not ameliorate ER stress or Hsp90 inhibition or non-proteasome flux stresses. In cells with reduced levels of PSMD2, bortezomib treatment strongly repressed genes involved in DNA replication. This heightened anti-proliferation response suggests that cells with reduced 19S subunits are primed to enter a protected, quiescent-like state when flux through the proteasome is reduced. Moreover, reducing PSMD2 levels maintained proteasome degradation flux upon bortezomib treatment and maintained the normal rate of protein synthesis. These phenotypes in mammalian cells are preserved in yeast, identifying a mechanism preserved over a billion years of evolution. In cancer cell lines, proteasome resistance is often coupled to reduction in the expression of one or more genes encoding the 19S subunits of the proteasome, speaking to significance. The central questions I have are:

1) How is it that a reduction of 19S regulatory subunit of the proteasome would render the cells less sensitive to inhibition by proteasome inhibitors?

2) How the 20S subunit increases relative to the 26S subunit is not clear to me mechanistically; please address.

*Reviewer #2*:

This is a very interesting paper that makes the surprising observation that knockdown of subunits of the proteasome 19S cap renders cells moderately resistant to CFZ. This is a very counter-intuitive observation that could be of considerable importance as it sheds new light on regulatory mechanisms that enable cells to deal with proteotoxic stress.

Overall, I find this to be worthy of serious consideration for publication in *eLife.* My main concern has to do with the hypothesis put forward by the authors to explain why cells with diminished 19S cap are more resistant to bortezomib and/or MG132. The chief argument appears to be that a subset of the 20S is refractory to inhibition in these cells. However, there are some problems with the experiments that show this. In the first experiment (Figure 2 panel F), there is indeed less 20S activity in WT cells treated with bortezomib compared with 19S-deficient cells, However, there is also much less proteasome. Where is the proteasome going in the WT control? Is this simply a consequence of cell death? But compare with panel 2B – cell number actually seems to be increasing over the first 24h. This experiment would benefit from being repeated as a time-course, so that it is possible to evaluate proteasome activity at a time before the proteasome has largely disappeared from the WT cells. It would also be much better to do this experiment with epoxomicin or carfilzomib because BTZ can potentially dissociate during the native gel electrophoresis, given that the t1/2 has been reported to be in the range of 100 minutes. Finally, it would be desirable to see this effect quantified (e.g. with an LLVY-AMC assay or similar). In Figure 4 panel C, a similar experiment is done with transiently depleted cells. Unlike Figure 2, there is not dramatic loss of proteasome in 19S-depleted cells. However, there is essentially undetectable level of 20S activity in the cells treated with BTZ. A time-course and/or dose-titration would be informative. Together, Figures 2 and 4 are key results for the paper, but the existing data are just not all that compelling.

The second major point is that I would have liked to see more of a discussion regarding possible mechanisms. How is it that a reduction of 19S would render the cells less sensitive to inhibition by bortezomib? There are several possible explanations, some of which are testable, but the authors are relatively silent on this salient point.

---

## [Author Response]

We have added additional experiments as requested (and also one that was not requested but further clarifies our points). We have also made modifications to the text. We believe we have thoroughly addressed all concerns and hope you will agree.

In the original version of the manuscript, we failed to make an important point clear: the increase in 20S proteasome complexes occurs in cells with reduced 19S subunits without an increase in the expression of the individual 20S subunit mRNA or protein levels. We have rearranged the panels in Figure 2 to better highlight this point. There are sharp changes in 20S complex levels in the native gel panels while total subunit levels remain unchanged in the corresponding SDS PAGE gels (Figure 2).

We share the reviewers’ interest in the decrease in 20S proteasome complexes in control cells following bortezomib treatment (Figures 2 and 4). As the reviewer points out, the decrease is not due to cell death. The analysis had been done at 24 hours after the addition of bortezomib, a time point when cells retained full viability (Figure 2). The text and the figure legend now make this point more explicitly. In addition, as the reviewer suggested, we performed a time course, monitoring 20S proteasome complexes (native gel) and total subunit levels (SDS Page gel) at 3, 6 and 18 hours after exposure to bortezomib (Figure 4—figure supplement 1). A significant decrease in the 20S proteasome *complex* occurred by 6 hours, but again, protein *subunit* levels were stable. Notably, the 19S subunit knockdown preserves the levels of 20S proteasome complexes throughout the bortezomib treatment (Figure 4—figure supplement 1).

We did not suggest that a subset of 20S proteasome complexes is inherently refractory to proteasome inhibition after 19S subunit knockdown. This is certainly an interesting possibility that is now mentioned in the text. It may simply be that cells with 19S knockdown start with such high levels of 20S complex that a significant fraction remains active at the concentrations of bortezomib that we used. To further investigate the increase in 20S complex levels and activity, we fractionated cell lysates using glycerol gradients with and without 19S subunit reduction and with and without bortezomib treatment (new Figure 4). The analysis of gradient fractions for proteasome subunit content and activity support our observations from native gel analysis and from the protein degradation assay (Figures 2 and 4). The 19S subunit reduction increases the 20S proteasome levels and activity and preserves proteasomal function in the presence of bortezomib (Figure 4).

As per the reviewers’ suggestion, we have included an additional paragraph in the Discussion describing additional possible mechanisms for our observations.